SOFTWARE

# RMPJ: An ImageJ plugin for morphological information processing in biomedical images

Yoshitaka Kimori *

Center for Mathematics and Data Science, Gunma University, Maebashi, Gunma, Japan

* kimori@gunma-u.ac.jp

## Abstract

RMPJ is an image-processing tool that facilitates the effective enhancement of low-contrast analysis objects and enables preprocessing to quantify the morphological features of the objects under analysis. This ImageJ plugin implements seven image-processing algorithms based on rotational morphological processing (RMP), a variation of traditional morphological operations. RMPJ enables the enhancement and extraction of intricate and nuanced features in biomedical images. Furthermore, it facilitates the separation and extraction of multiple aggregated structures. This software aims to expand the scope of research in life sciences by providing an image-processing method that quantitatively analyzes the morphological information of objects.

## Introduction

In the life sciences, numerous microscopes and medical modalities are utilized to visualize the structures of research objects and accumulate image data. The acquisition of considerable image information facilitates an in-depth understanding of the structure of the object under investigation. Image processing, involving contrast enhancement, denoising, and segmentation, is crucial in studying image information. The enhancement and denoising of images improve the clarity of the object under study, whereas segmentation extracts the object of analysis from the image.

Methods based on mathematical morphology are commonly used to process biomedical image data. Mathematical morphology provides nonlinear image processing techniques for extracting and analyzing useful shape information from digital images [1–3]. The mathematical morphology system performs set operations between the input image and a small image called a structuring element (SE). As a "probe," the SE scans the input image pixel by pixel and applies the operations on each local region of the image, resulting in a new processed image.

The basic operations are dilation and erosion, and a combination of these can create various morphological operations. Opening is an operation that applies the erosion to the original image, followed by the dilation to the eroded image. Its dual

**Data availability statement:** The RMPJ code is available on GitHub at https://github.com/ykimori/rmpj.

**Funding:** Work on this paper by YK was supported by JSPS KAKENHI Grant Number JP22H04926, Grant-in-Aid for Transformative Research Areas – Platforms for Advanced Technologies and Research Resources "Advanced Bioimaging Support". The funders had no role in study design, data collection and analysis, decision to publish, or preparation of the manuscript.

**Competing interests:** The author have declared that no competing interests exist.

operation is called closing. Closing involves the application of dilation to the original image and erosion to the dilated image. Opening removes structures smaller than the SE and closing fills structures such as holes and grooves smaller than the SE in the object region. In contrast, both operations preserve the regions where the SE is fitted. Top-hat transforms are operations that extract the structure removed (resp. filled in) by opening (resp. closing) as a meaningful feature. Two types of top-hat transform exist: white top-hat, which subtracts the opened image from the original image, and black top-hat, which subtracts the original image from the closed image (for details, see Section 1.1. of S1 Text).

However, when applying traditional morphology operations to biomedical images, the fine and complex structure of an object to be analyzed may be distorted or destroyed depending on the shape of the SE and the orientation of the object.

SEs are represented discretely. Therefore, even if the SE is disk-shaped, its discretized shape is not invariant with respect to rotation at arbitrary angles. Thus, isotropic processing cannot be achieved for the object being analyzed in morphological processing using this SE. This may cause damage to the shape of the object being analyzed.

Depending on the orientation of the object being analyzed, the SE may not fit with the structure to be preserved, resulting in its removal and deformation of the object's overall shape.

For example, the opening operation is commonly used to remove noise from images. Although it is effective in noise removal, it can also distort the shape of the object being analyzed. In biomedical research, image processing is primarily used to quantify the morphological features of the object being analyzed. Therefore, any deformation of the object by image processing can significantly affect the results of the morphological feature analysis.

Additionally, when processing biomedical images, extracting low-contrast granular objects from backgrounds with uneven brightness is often required. However, traditional morphological processing struggles to meet this requirement with sufficient accuracy for analysis, such as measuring particle number or spatial distribution.

To address these problems, a more resilient and adaptable computational method called rotational morphological processing (RMP) was developed. The principle of its algorithm is that the original image is rotated from various angles, and (traditional) morphological processing is performed after each rotation. The same SE is used for this processing. The images to which morphological processing has been applied are then combined into a single image and output (for details, see Section 1.2. of S1 Text).

Even when using the aforementioned discretized SE, RMP-based operations can be similar to isotropic processing. By rotating the image at various angles, RMP preserves the overall shape of the analyzed object, regardless of its orientation.

Furthermore, the requirement to extract granular objects can be achieved using RMP-based top-hat transform with line-shaped SE. A key feature of RMP is its ability to fully leverage the line-shaped SE, which has traditionally been limited to specific applications in mathematical morphology. This approach enables the extraction of closely spaced granular objects that cannot be separated using other SEs.

Image processing based on RMP has been employed in various studies, including protein structural analysis using electron microscopy [4], spot-like (granular) object extraction in fluorescence microscopy images [5], morphological analysis of intracellular organelles in plant cells [6], and lesion enhancement in medical images [7,8], to achieve research objectives.

However, researchers in the biological and medical fields with limited programming expertise face difficulties utilizing RMP-based image processing algorithms owing to the lack of reliable and user-friendly implementations. Several software packages, including MATLAB, incorporate traditional morphological processing, and ImageJ [9] plugins [10] are also available. However, they do not implement RMP-based image processing algorithms.

Hence, to mitigate the limitations of existing software, this study designed a novel plugin for ImageJ, an image processing and analysis tool used in various scientific research fields. The designed plugin, RMPJ, implements algorithms based on RMP. It enhances and extracts intricate features in biomedical images, enabling the separation and extraction of multiple aggregated structures.

RMPJ incorporates seven morphological operation algorithms based on the RMP developed by the author. It integrates with various ImageJ functions and features a user-friendly graphical user interface (GUI).

RMPJ has the potential to be a valuable resource for research groups facing difficulties that cannot be solved by conventional image processing techniques, such as those associated with enhancing low-contrast objects or the development of preprocessing methods to quantify the morphological features of the analyzed object.

The theoretical background of the mathematical morphology and RMP is detailed in S1 Text.

## Design and implementation

RMPJ is a Java-based ImageJ plugin that applies RMP to single and stacked images. The plugin consists of seven RMP-based operators: opening, closing, white top-hat, black top-hat, morphological smoothing, morphological contrast enhancement type 1, and morphological contrast enhancement type 2.

Opening (resp. closing) serves the function of removing (resp. filling in) structures smaller than the SE, while preserving structures that fit the SE. It is used to remove noise, among others. White top-hat (resp. black top-hat) is a filter that extracts and emphasizes structures that have been removed (resp. filled in) by opening (resp. closing) operations. Morphological smoothing is a smoothing filter that combines opening and closing operations. Morphological contrast enhancement type 1 filter extracts fine structures by subtracting the smoothed image from the original, enhancing these structures. Morphological contrast enhancement type 2 is a filter that simultaneously emphasizes the structures extracted by white top-hat and black top-hat. For details of these operations, see Section 1.2. of S1 Text.

The RMPJ plugin allows to setup four input options:

(1) Operation type: Seven operators—Opening, Closing, White top-hat, Black top-hat, Smoothing, Enhance-type1, and Enhance-type2—can be selected from a dropdown list. These implement operators $\gamma^R$(Eq (S11)), $\phi^R$(Eq (S12)), $WTH^R$ (Eq (S13)), $BTH^R$(Eq (S14)), $MS^R$(Eq (S15)), $MCE_1^R$(Eq (S16)), and $MCE_2^R$(Eq (S17)), respectively.

(2) Shape of structuring element: Three types of SEs, Disk, Line, and Square, can be selected to define the shape of the structuring element.

(3) Size of structuring element: The size of the SE can be defined in pixels. For a disk-shaped SE, the diameter is used; for a square-shaped SE, the length of one of its sides is used; and for a line-shaped SE, the length of the line is used.

(4) Number of image rotations: This number ($N$) refers to the number of times the image is rotated. A value according to the shape of the SE used and the purpose of the processing is entered. The optimum value of $N$ is 8 when a square or disk-shaped SE is used, and the optimum value of $N$ is 36 when a line-shaped SE is used.

When $N$ = 1, the angle of rotation is set to 0°. In this case, traditional morphological operations are performed.

The results of the evaluation experiments, which provide guidelines for determining the shape of the SE and the value of *N,* are shown in Section 2.4. of S1 Text.

By selecting the "Stack options" checkbox, the user can specify the processing of all images in the stack, only the images within a specific range in the stack, or only the selected images.

These operations are classified into two groups: image smoothing and image enhancement. Opening, Closing, and Smoothing can be used as image processing filters with smoothing functions. White top-hat, Black top-hat, Enhance-Type1, and Enhance-Type2 can be used as image enhancement filters.

The image enhancement algorithms underlying Enhance-Type1 (explained in detail in Reference [7]) and Enhance-Type2 (explained in detail in References [8]) are less popular and rarely used compared to top-hat transforms. However, they possess features absent in top-hat transforms, making them effective for emphasizing fine structures with extremely low contrast against the background.

The plugin software can process 8-bit and 16-bit grayscale images. Current specifications limit the maximum size of the input image to 2048×2048 pixels and the maximum size of the SE to 99×99 pixels.

In this software plugin, image histogram modification can be performed on the results of the top-hats and two image enhancement filters. If the image has a negative intensity value, the dynamic range is shifted such that the minimum value of the image is set to zero. Subsequently, a histogram equalization process is applied to the resulting image to achieve a uniform distribution of the image histogram. Finally, a linear contrast stretching process is applied to the equalized histogram to enhance the image dynamic range. This process can be used to produce contrast-enhanced images.

The software employs the bilinear method for image rotation interpolation.

## Results

This section demonstrates the efficacy of RMPJ through morphological image-processing examples. Specific procedures for image processing with ImageJ are described in S2 Text.

### Effect of the different types of SE shapes and the number of image rotations on the processing results

$\gamma^R$ was executed using square- and disk-shaped SEs. The results were compared for $N = 1$ and $N = 8$. When $N = 1$, the image was not rotated, and the $\gamma^R$ result was identical to that of a traditional opening operation.

The upper image in Fig 1a is the original image (128×128 pixels). This image is artificially synthesized by placing a mesh structure, followed by a circular structure in the background, and then adding noise to the entire image. We consider the extraction of the mesh structure from this image. The lower image in Fig 1a depicts the optimal outcome of the structure extraction process. This image is used as the ground truth for the mesh structure. An opening operation was applied to eliminate noise and unwanted structures surrounding the mesh structure in the original image. The upper images in Fig 1b and 1c depict the outcome of $\gamma^R$ using a square-shaped SE; the size of the SE is 7 pixels. Fig 1b and 1c show the results for $N = 1$ and $N = 8$, respectively. Otsu's automatic thresholding method [11] was applied to these images (see Section 1 in S2 Text), and the results converted to binary images are shown in the lower images in Fig 1b and 1c. The upper images in Fig 1d and 1e show the results of $\gamma^R$ with a disk-shaped SE (size: 7 pixels), Fig 1d shows the results for $N = 1$, and Fig 1e shows the results for $N = 8$. The lower images of Fig 1d and 1e show the results obtained by applying Otsu's method to the images.

These results indicate that the $\gamma^R$ outcome with a disk-shaped SE for $N = 8$ is the most similar to that for the ground truth. The mesh structure was extracted without interruption, and all unnecessary structures, such as noise, were removed.

A quantitative evaluation of the similarity between the results of these openings and the ground truth is shown in Section 2.4. of S1 Text.

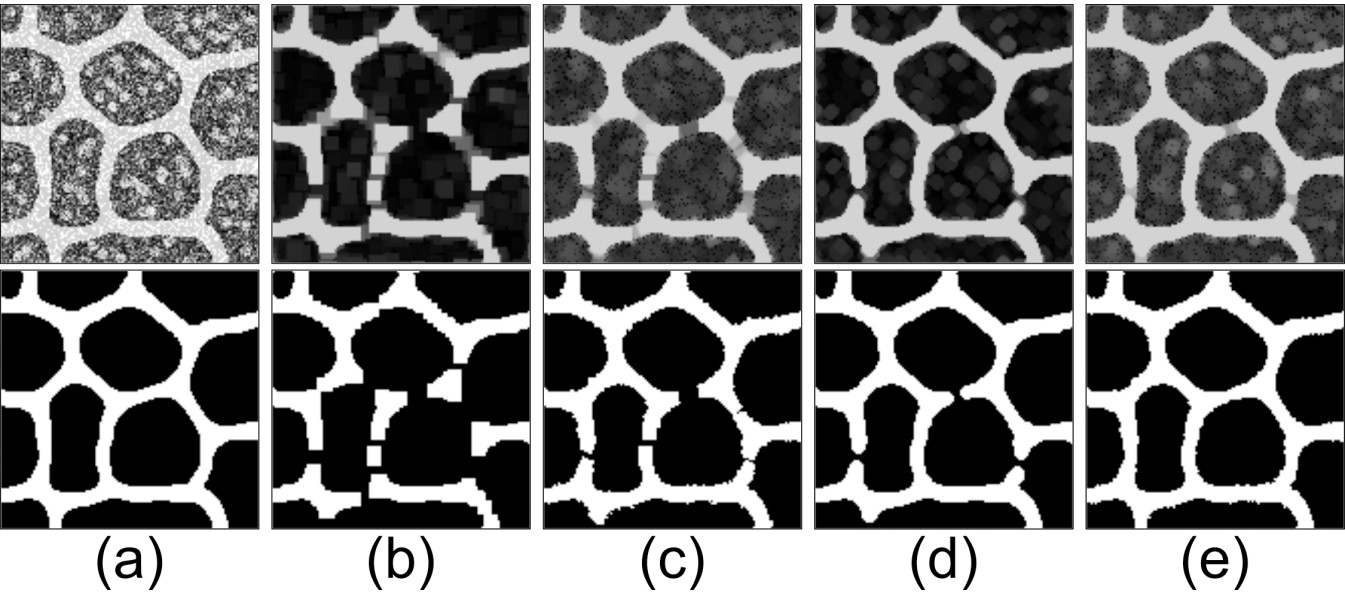

**Fig 1. Extraction of mesh structure based on smoothing by the opening operation.** (a) Original image (upper) and ideal extraction result for mesh structure (ground truth) (lower). The images in the upper rows from (b) to (e) are the result of applying $\gamma^R$. Conditions for SE shape and $N$: (b) square-shaped SE and $N = 1$; (c) square-shaped SE and $N = 8$; (d) disk-shaped SE and $N = 1$; and (e), disk-shaped SE and $N = 8$. The images in the lower rows of (b) through (e) are binarized images for the $\gamma^R$ results shown in the upper row of each image.

### Microstructure enhancement in medical images: Differences in image enhancement effects among the three image processing methods

This section presents the differences among the effects of the $WTH^R$, $MCE_1^R$, and $MCE_2^R$ methods on the enhancement of tissue microstructures in mammographic images. Fig 2a depicts a mammographic image with a region containing a mass lesion (indicated by an arrow) that is enlarged and shown in the lower panel. The original image is from mdb021ll in the Mammographic Image Analysis Society (MIAS) database v1.21 [12]. This image has been reduced to half of its original size, and the area containing the breast has been cropped from the entire image.

For all operations, a disk-shaped SE (size: 45 pixels) was used, and the value of $N$ was set to 8. Fig 2b displays the results of $WTH^R$, which extracts and enhances filament structures, such as mammary glands, regardless of the intensity of the original image. The region of the mass lesion is not emphasized (because this region is larger than the size of the SE); however, the overlapping filament structure is emphasized. Fig 2c depicts the results for $MCE_1^R$. The extracted image reveals the filament structures as well as the low-intensity regions in the original image (the groove regions between the filaments). The $MCE_2^R$ emphasizes not only the filament structures, but also the entire breast region. (Fig 2d). In particular, the boundary of the breast, which was difficult to observe, can be recognized (indicated by arrowheads).

Processing of filament-like structures is a primary application of RMP-based operations. In addition to enhancing low-contrast fine structures, as demonstrated in this section, RMP-based methods are also used to analyze the complexity and directionality of filamentous structures [6].

### Application to stacked image data

RMPJ can also handle stacked 2D image data. Each image in the stack is processed individually. This section presents an example of stacked data (mouse embryo blastocyst cells, BBBC032v1 from https://bbbc.broadinstitute.org/BBBC032) processing obtained from the Broad Bioimage Benchmark Collection [13]. Here, image stack data (BMP4blastocystC1.tif

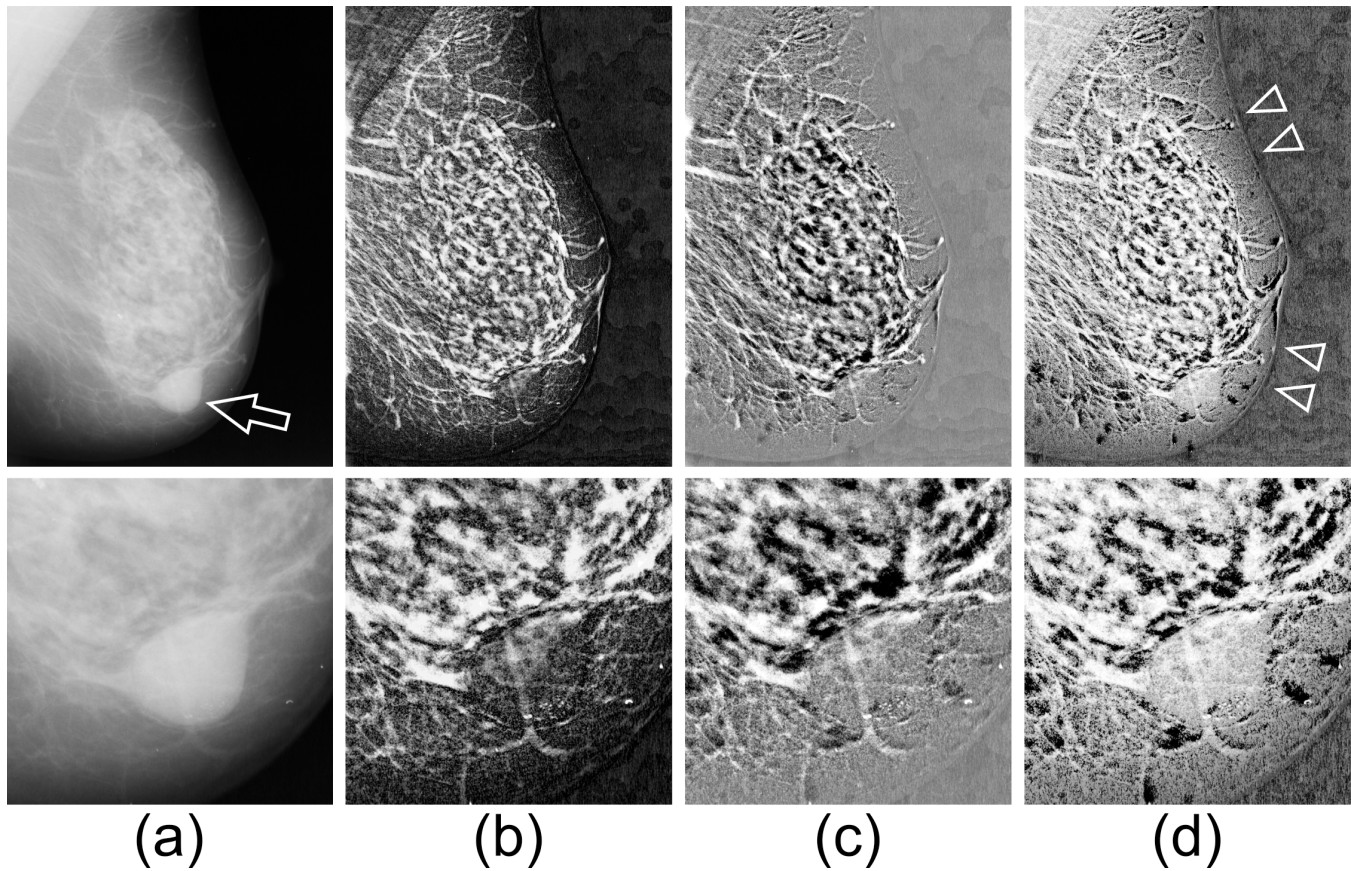

**Fig 2. Microstructure enhancement from the mammography image.** Original image (a), resultant images by (b) $WTH^R$, (c) $MCE^R_1$, and (d) $MCE^R_2$. The lower image in each row shows a magnified view of the region surrounding the mass lesion.

(Fig 3a), BMP4blastocystC2.tif (Fig 3b), and BMP4blastocystC3.tif (Fig 3c) obtained from three channels were used for processing. The top panels of Fig 3a through Fig 3c show the original images of each data stack. The images of slice numbers 110 and 145 in the stacked data are shown as examples.

The flow for processing each stack is as follows:

(1) The $x - y$ plane of the original data is reduced by half.

(2) The image intensity bit in the original data (16-bit) is converted to 8-bit.

(3) Gaussian blurring ($\sigma = 1$) is applied to reduce noise.

(4) $WTH^R$ is applied to enhance the target structures.

(5) To segment the $WTH^R$-enhanced structures, binarization is performed using Tsai's moment-preserving thresholding method [14].

These procedures using ImageJ are described in detail in Section 2 of S2 Text.

When processing BMP4blastocystC1.tif, $WTH^R$ with a line-shaped SE (size is 7 pixels) was applied to the enhanced aggregated fluorescent spots. When processing BMP4blastocystC2.tif, $WTH^R$ with a disk-shaped SE (size is 7 pixels) was applied to the enhanced cell membrane structures. When processing BMP4blastocystC3.tif, $WTH^R$ with a line-shaped SE

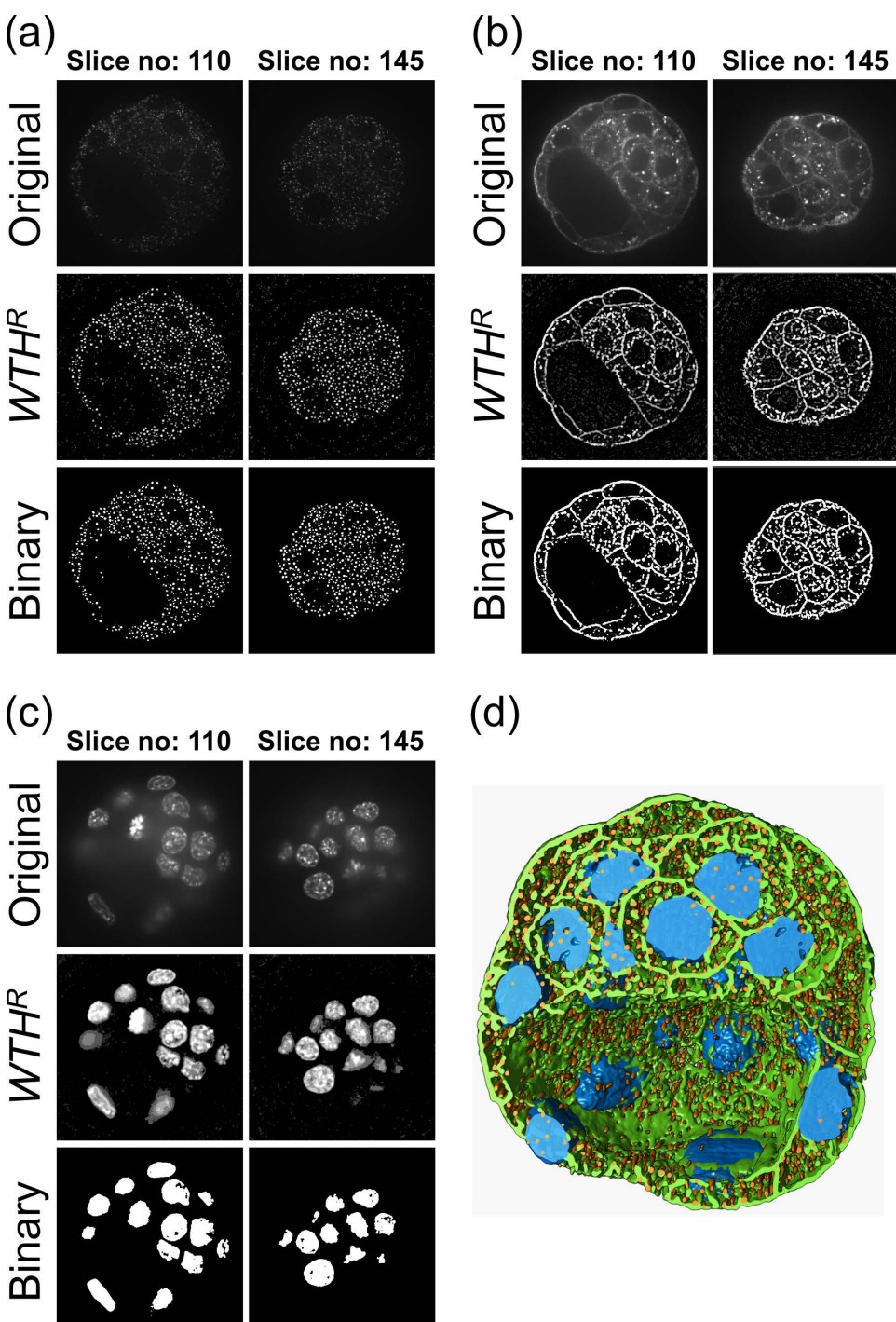

**Fig 3. Examples of stack image data processing.** The original images in (a) to (c) were obtained from the BBBC032v1 dataset. These images are those of slice numbers 110 and 145 in the stacked image data of BMP4blastocystC1.tif (a), BMP4blastocystC2.tif (b), and BMP4blastocystC3.tif (c). The fluorescent spots from BMP4blastocystC1.tif, the cell membrane structures from BMP4blastocystC2.tif, and the cell nuclei regions from BMP4blastocystC3.tif were enhanced by $WTH^R$, and their structures were segmented using a binarization method. (d) 3D reconstructed images of segmented structures.

(size is 95 pixels) was applied to enhance cell nuclei regions. These $WTH^R$ processing results are shown in the middle panels of Fig 3a through 3c. The binarized results of the $WTH^R$ images are shown in the bottom panels of Fig 3a through Fig 3c. These results show that the three types of structures enhanced by $WTH^R$ are segmented.

Fig 3d shows the results of 3D reconstruction using three types of binarized data (the results for BMP4blastocystC1.tif, BMP4blastocystC2.tif, and BMP4blastocystC3.tif are shown in orange, green, and blue, respectively). The 3D structure of a blastocyst cut in a certain section is displayed. For 3D reconstruction and visualization, UCSF Chimera [15] was used.

## Availability and future directions

RMPJ can be used to achieve the following three main objectives in the image processing of biomedical image data:

(1) Elimination of noise and artifacts that do not destroy or deform the structure of the object to be analyzed.

(2) Enhancement of the fine structure of the object to be analyzed.

(3) Separation and extraction of the agglomerated particles.

Objective (1) can be achieved using operations $\gamma^R$, $\phi^R$, and $MS^R$. By rotating the image, the SE can be fitted to structures of interest in various orientations; for example, in the opening operation, the regions where the SE fits are preserved, and regions where the SE does not fit (i.e., noise or artifacts smaller than the size of the SE) are removed. Objective (2) can be achieved using the $WTH^R$, $BTH^R$, $MCE_1^R$ and $MCE_2^R$ operators. In these operations, the structure to be emphasized can be selected by changing the size and shape of the SE. Objective (3) can be achieved through operations such as $WTH^R$ using a line-shaped SE. This objective is challenging to achieve with other image processing methods, such as traditional morphological operations. A key feature of RMP is its ability to make effective use of the line-shaped SE. These processes are essential for a quantitative understanding of the morphology of the analyzed object.

In addition, multiple analysis targets with varying shapes can be selectively extracted from the processed data using multiple SEs that correspond to the size and shape of each structure to be extracted.

Because RMPJ is a plugin for ImageJ, it can be used in conjunction with ImageJ functions for processing before (such as applying a Gaussian blur) and after (such as applying binarization using the automatic thresholding method and measuring object shape) the RMP application.

The opportunities for using deep learning for image recognition tasks in biomedical research are expected to grow in the future. RMPJ could be used as a tool for generating high-quality training data for deep learning. Deep learning necessitates a substantial amount of training images, which may require the difficult task of extracting target regions. RMPJ can automate most of this task, potentially reducing the workload. By using RMPJ as an image preprocessor to remove background structure and noise, which can degrade the object recognition performance of neural network models, higher-quality training data can be achieved [16].

Path operators [17–19] and geodesic transformations [3] are mathematical morphology theories with the same purpose and direction as the RMP approach, whereas pattern spectrum [20] is a mathematical morphology application method with the potential to introduce RMP-based operations. In the future, I plan to conduct comparative research between these theories and the RMP theory, as well as attempt to introduce RMP into existing methods. These studies will expand the functionality of RMPJ.

RMPJ is freely available on https://github.com/ykimori/rmpj.

## Supporting information

**S1 Text. Theoretical background of mathematical morphology and rotational morphological processing.**
(PDF)

**S2 Text. Image processing procedure using ImageJ.**
(PDF)

## Author contributions

**Conceptualization:** Yoshitaka Kimori.

**Data curation:** Yoshitaka Kimori.

**Formal analysis:** Yoshitaka Kimori.

**Funding acquisition:** Yoshitaka Kimori.

**Investigation:** Yoshitaka Kimori.

**Methodology:** Yoshitaka Kimori.

**Project administration:** Yoshitaka Kimori.

**Resources:** Yoshitaka Kimori.

**Software:** Yoshitaka Kimori.

**Validation:** Yoshitaka Kimori.

**Visualization:** Yoshitaka Kimori.

**Writing – original draft:** Yoshitaka Kimori.

**Writing – review & editing:** Yoshitaka Kimori.

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
