## [Decision Letter · Decision Letter 0]

24 Sep 2024

Dear Prof. Kimori,

Thank you very much for submitting your manuscript "RMPJ: An ImageJ plugin for morphological information processing in biomedical images" for consideration at PLOS Computational Biology.

We appreciate your patience as we tried to secure an editor and to receive reviewer comments.

As with all papers reviewed by the journal, your manuscript was reviewed by members of the editorial board and by several independent reviewers. In light of the reviews (below this email), we would like to invite the resubmission of a significantly-revised version that takes into account the reviewers' comments.

In particular, the reviewers suggest a better introduction and details regarding RMP-based image processing. Additionally, the reviewers request other details about the approach and its limitations of the approach. These comments should be addressed in the resubmission.

We cannot make any decision about publication until we have seen the revised manuscript and your response to the reviewers' comments. Your revised manuscript is also likely to be sent to reviewers for further evaluation.

Sincerely,

Stacey D. Finley, Ph.D.

Section Editor

PLOS Computational Biology

Reviewer's Responses to Questions

**Comments to the Authors:**

Reviewer #1: This paper clearly describes that the image processing method of rotational morphological processing (RMP) is widely applicable to biomedical research fields, which is supported by the author's previous work cited in the references. The explanation of RMP described in S1 Text is clear and understandable. The publication of this technique (RMPJ) is likely to accelerate a lot of research. Therefore, I recommend this paper for publication.

The following are some minor questions and comments for the author to consider when polishing the final manuscript. (Discretionary revisions)

1) How was the original image in Fig. 2a produced? (Original samples used in other figures are clearly explained).

2) Some things and parameters need to be determined to perform RMP-based processing, shape of the structuring element (SE), its size, number of iterations, etc. Does the author have any guidance on how to achieve a correct result other than trial and error?

3) The author demonstrates a good application result for stacked image data. For such serial image data, does the author think there is a need to extend the 2D SE to a 3D SE (voxel type) in the future?

Reviewer #2: Review for the manuscript "RMPJ: An ImageJ plugin for morphological information processing in biomedical

images", by Yoshitaka Kimori.

The mansucript presents a software devoted to Rotational Mathematical Morphology for ImageJ.

The interest of Mathematical Mrophology (MM) is first introduiced, then the software is presented.

Several application examples are presented as use cases of the software.

The concept of rotational morphology Processing (RMP) is clearly an interesting approach, as it can

help in many situations. To see it implemented in a user-friendly plugin for ImageJ is a great advance.

There are however some issues in the proposed manuscript that should be adressed. My main concern is

about the general introduction of the concepts, that makes the reading much difficult that it could be.

More specifically, it would be much more clear if RMP was more explained in the introduction.

I would typically expect some examples of limitations of non rotational MM, then a presentation of

the general concept of RMP (refering to S1 for mathematical details), and finally the expected advantages.

The introduction of new operators could also be emphasized in the introduction.

There are also some technical concerns in the supplementary S2.

I understand there should be some space limitations, but in some places, the text could be shorten without

losing information.

I have also some remarks, that could be adressed in the discussion:

* how does RMP relate to path opening methods developed by Heijmans et al.? It seems to me there may be

similar objectives.

* From a technical point of view, would it make a big difference to apply the rotation to the SE

instead of to the image? I suppose this is a question of discretisation issue?

* Would it be possible to extend the method to the processing of 3D images by applying 3D rotation?

* finally, the proposed RMP framework looks promising. Beyond the presentation of the RMPJ software,

I wonder is a methodological article could have been more appropriate?

More specific comments:

L45: maybe pixel by pixel is more appropriate?

L47: a quick explanation of what are opening, closing and top-hats would make the reading more comfortable.

L72: the software propose rotational version of classical operators, as well as new operators (MCE1 and MCE2).

This can be better emphasized.

L90: It would be nice to give a short explanation of what do the different operators do.

Typically in the introduction, or in the beginning of the Design section.

L103: I am surprised that an optimum value for line SE exist. I would have expected it to depend on the application,

typically on the curvature or thickness of the structures to filter.

L124-127: part of this paragraph could be presented in the introduction.

Fig.2: I am surprised that the result of WTH and WTH^R with disk differ. As the disk is

(in continuuous domain at least) rotation invariant, I would expect to obtain the same result.

L169+: again, I do not understand the interest of using RMP with a disk structuring element?

L246: GitHub link is not active...

S1: the MCE1 and MCE2 operators seems to be new. I suggest to present them more in details in the main part of manuscript.

S1, section 1.2: I strongly disagree with the sentence "the constant scanning direction of the SE

during the morphology operation causes the deformation of the shape of the object."

The scanning order has nothing to do with the deformation of the objects. The deformation is caused solely by

the shape of the SE. The same result is obtained when scanning the image backwards, or in random order.

Please correct.

S1. I am surprised not to see an example based on the enhancement of filament-like structure, similar to what was

presented in Kimori et al, 2016. To my opinion, this is one of the main use-cases of the proposed method!

**Have the authors made all data and (if applicable) computational code underlying the findings in their manuscript fully available?**

Reviewer #1: Yes

Reviewer #2: Yes

PLOS authors have the option to publish the peer review history of their article (what does this mean? ). If published, this will include your full peer review and any attached files.

**Do you want your identity to be public for this peer review?** For information about this choice, including consent withdrawal, please see our Privacy Policy .

Reviewer #1: No

Reviewer #2: No
---

## [Decision Letter · Decision Letter 1]

26 Jan 2025

PCOMPBIOL-D-24-00594R1

RMPJ: An ImageJ plugin for morphological information processing in biomedical images

PLOS Computational Biology

Dear Dr. Kimori,

Thank you for submitting your manuscript to PLOS Computational Biology. After careful consideration, we feel that it has merit but does not fully meet PLOS Computational Biology's publication criteria as it currently stands. Therefore, we invite you to submit a revised version of the manuscript that addresses the points raised during the review process.

Please submit your revised manuscript within 60 days Mar 28 2025 11:59PM. If you will need more time than this to complete your revisions, please reply to this message or contact the journal office at ploscompbiol@plos.org. Please include the following items when submitting your revised manuscript:

We look forward to receiving your revised manuscript.

Kind regards,

Stacey D. Finley, Ph.D.

Section Editor

PLOS Computational Biology

**Additional Editor Comments :**

Reviewer 2 has raised additional points that are important to address, including the use of the disk shape and references to prior work in this area.

**Journal Requirements:**

1) Please provide an Author Summary. This should appear in your manuscript between the Abstract (if applicable) and the Introduction, and should be 150-200 words long. The aim should be to make your findings accessible to a wide audience that includes both scientists and non-scientists. Sample summaries can be found on our website under Submission Guidelines:

**Reviewers' comments:**

Reviewer's Responses to Questions

Reviewer #1: There are no specific comments on the revised manuscript.

Reviewer #2: Review for Kimori Revision 1

First of all I thank the author for the updates on the manuscript, and for the detailed

responses to the comments. They help clarify the presentation and make it easier

to understand several points. I also took more time to investigate previous works on related topics.

My global opinion is ambivalent:

* I strongly support the development and the promotion of a plugin that allows rotational

mathematical morphology for the (widely used) software ImageJ!

* However I have a strong concern with the notion of rotation morphological processing

using disk-shaped structuring element...

To be more precise, is not clear to me which effect is implied For RMP with disk.

The global workflow involves two rotations (before and after applying morphological operation).

In addition to the morphological operation, there is therefore a smoothing effect due to the

interpolations. My feeling is that the results presentged for disk are an artifact obtained

from the resampling of the rotated image, even if I could not find a way to demonstrate it...

Also, while linear interpolation is the most classical choice, other methods could ne chosen,

possibly leading to different results.

Anyway, relying on such an effect for presenting the software, or more generally to use as basis

the differences in the discretisations of a disk, seems to me highly discutable.

Moreover it may lead to controversies about the general validity of the RMP method/framework,

that would be detrimental to the presented software.

However, I am strongly convinced that the RMP method is very useful and well adapted for non-disk

structuring elements.

I strongly encourage to revise the manuscript, by putting more emphasis on square and linear SE,

and removing the presentation of RMP based on disks.

I also feel there is room for further investigations on the links between RMP, path-opening, and

related methods... Even if larger comparison is out of the scope of the manuscript, it would be

beneficial to add few references to link to related works (for example in the introduction).

Few examples include:

* P. Soille, H. Talbot, Directional morphological filtering, IEEE Trans. Pattern Anal. Machine Intell., 23 (11) (2001), pp. 1313-1329

* H. Talbot, B. Appleton, Efficient complete and incomplete path openings and closings, Image Vision Comput., 25 (4) (2007), pp. 416-425

* Merveille, O.; Talbot, H.; Najman, L. & Passat, N. Curvilinear Structure Analysis by Ranking the Orientation Responses of Path Operators

IEEE Transactions on Pattern Analysis and Machine Intelligence, 2018, 40, 304-317

There are also examples in the book of Soille.

more minor point, I would advocate of avoiding the "iterations" term. Using the term "iteration"

may lead the reader that the algorithms updates its internal state at each iteration, such that

each iteration depends on the previous one. Typical examples are k-means or gradient descent.

In this case, it seems the term 'rotation angle' could be used more adequately?

Minor remarks

comment 2:

I have discussed the point above; The figure R1 in the response to comment 2 is surprising,

as the square did not rotate around its center.

Also, be careful about sentence "rotating the image without any interpolation" -> there *was*

an interpolation, but in this case it was "nearest-neighbor interpolation".

comment 3:

I totally agree with the response. Extension to 3D may be envisoned, but would be complicatd.

comment 4:

I took more time to dig into the previous works, and I better understand the objective of the paper.

It is totally fine to choose a software paper.

comment 7:

I did not fully realize that methods MCE1 and MCE2 were already presented, thank you for clarifying it!

comment 9:

Well, I still consider that the number of rotations should depend on the length or size of

the structuring elements, but ok for me.

comment 11:

Thank you for the detailed explanation and figure.

I am sorry to say that this does not help understand the principles that makes

the RMP framework seem to work... As discussed previsouly, I would "incriminate"

interpolation artifacts.

comment 12:

Again, thank you for the detailed explanations and figure.

I have the impression that a classical top-hat with a disk (not rotational) would

have the same result. This enforce me in the conviction to avoid putting the emphasis

in RMP with disk.

comment 13:

Thank you for making the development repository publicly available. Making development open-source

grealty favor diffusion of methods and tools!

Note also that the GitHub platform allows for generating "releases", making it convenient

to distribute the packaged jar archive.

L59: Some authors include the terms "resp." within parens to make explicit that the term

within parens is the alternative case as the one in plain text.

https://math.stackexchange.com/questions/3826526/what-this-resp-means

L66: This paragraph is ambiguous and discutable. If the SE is a (contiunous) disk,

it *is* rotation invariant, and its discretisation will remain the same.

However, What I understand from the paragraph, is that the *discretisation* of the SE

is not invariant to rotation. At least add "discretized" before "shape" would be enough to clarify.

L86: remove "iteratively". while the method uses a for-loop, this is not an iterative algorithm

that updates its state after each iteration.

L137: I would rephrase as "The RMPJ plugins allows to setup four input options:"

L148: I would replace by "the number of image rotations to consider" (one may think that

rotations are applied in successions)

L186: I am still convinced that for this example it would be more appropriate to use line structuring elements...

L187: same as L86; can be simplified in "... and the number of image rotations on ..."

**Have the authors made all data and (if applicable) computational code underlying the findings in their manuscript fully available?**

Reviewer #1: Yes

Reviewer #2: Yes

PLOS authors have the option to publish the peer review history of their article (what does this mean? ). If published, this will include your full peer review and any attached files.

**Do you want your identity to be public for this peer review?** For information about this choice, including consent withdrawal, please see our Privacy Policy .

Reviewer #1: No

Reviewer #2: No

**Figure resubmission:**
---

## [Decision Letter · Decision Letter 2]

24 Mar 2025

Dear Prof. Kimori,

We are pleased to inform you that your manuscript 'RMPJ: An ImageJ plugin for morphological information processing in biomedical images' has been provisionally accepted for publication in PLOS Computational Biology.

Best regards,

Stacey D. Finley, Ph.D.

Section Editor

PLOS Computational Biology

Reviewer's Responses to Questions

**Comments to the Authors:**

Reviewer #2: First of all I would like to thank the author for the numerous efforts in answering the remarks

and providing additional results for illustrating the different points that were raised.

This really help understanding the point of view that is proposed.

I still have interrogation about the mathematical bases of the methods, but I think this goes beyond

the scope of the review process. Moreover, the proposed software is valuable and will be of great

beneficial to the (bio) image analysis community. The minor remarks have also been answered.

Therefore, I agree to make the manuscript published in the present form.

**Have the authors made all data and (if applicable) computational code underlying the findings in their manuscript fully available?**

Reviewer #2: Yes

PLOS authors have the option to publish the peer review history of their article (what does this mean? ). If published, this will include your full peer review and any attached files.

**Do you want your identity to be public for this peer review?** For information about this choice, including consent withdrawal, please see our Privacy Policy .

Reviewer #2: No

---

## [Editor Report · Acceptance letter]

PCOMPBIOL-D-24-00594R2

RMPJ: An ImageJ plugin for morphological information processing in biomedical images

Dear Dr Kimori,

I am pleased to inform you that your manuscript has been formally accepted for publication in PLOS Computational Biology. Your manuscript is now with our production department and you will be notified of the publication date in due course.

With kind regards,

Anita Estes
